# Optimization of Ultrasonic-Assisted Extraction Conditions for Bioactive Components and Antioxidant Activity of *Poria cocos* (Schw.) Wolf by an RSM-ANN-GA Hybrid Approach

**DOI:** 10.3390/foods12030619

**Published:** 2023-02-01

**Authors:** Shiqi Chen, Huixia Zhang, Liu Yang, Shuai Zhang, Haiyang Jiang

**Affiliations:** Department of Veterinary Pharmacology and Toxicology, College of Veterinary Medicine, China Agricultural University, Beijing Key Laboratory of Detection Technology for Animal−Derived Food Safety, Beijing Laboratory for Food Quality and Safety, Beijing 100193, China

**Keywords:** *Poria cocos* (Schw.) wolf, artificial neural network, genetic algorithm, optimization, antioxidant

## Abstract

In this study, a response surface methodology and an artificial neural network coupled with a genetic algorithm (RSM-ANN-GA) was used to predict and estimate the optimized ultrasonic-assisted extraction conditions of *Poria cocos*. The ingredient yield and antioxidant potential were determined with different independent variables of ethanol concentration (X_1_; 25–75%), extraction time (X_2_; 30–50 min), and extraction solution volume (mL) (X_3_; 20–60 mL). The optimal conditions were predicted by the RSM-ANN-GA model to be 55.53% ethanol concentration for 48.64 min in 60.00 mL solvent for four triterpenoid acids, and 40.49% ethanol concentration for 30.25 min in 20.00 mL solvent for antioxidant activity and total polysaccharide and phenolic contents. The evaluation of the two modeling strategies showed that RSM-ANN-GA provided better predictability and greater accuracy than the response surface methodology for ultrasonic-assisted extraction of *P. cocos*. These findings provided guidance on efficient extraction of *P. cocos* and a feasible analysis/modeling optimization process for the extraction of natural products.

## 1. Introduction

*Poria cocos* (Schw.) Wolf (Fu-ling) is an edible fungus commonly found on the dead bark and roots of pine trees [1] which are distributed all over the world. To date, numerous scientific studies have revealed the bioactivities of *P. cocos*, such as its antioxidant, immunomodulator, anti-inflammatory, anticancer, and antidiabetic effects [2,3]. Additionally, in phytochemical analysis, many important functional components, mainly including triterpene acids (pachymic acid, trametenolic acid, tsugaric acid A, and dehydrotrametenolic acid), polysaccharides, and polyphenols, have been discovered in Fu-ling [4,5,6]. Nowadays, with its marked health benefits, non-toxicity, and rich resources, *P. cocos* or its extracts have been widely used as raw materials or supplements to produce beverages, porridge, biscuits, and functional foods [7,8].

Extraction processes are the critical step in the separation, purification, and identification of functional components in food raw materials [9]. Ultrasonic-assisted extraction is one of the most commonly used techniques, which can fully release the ingredients in raw materials by ultrasonic energy [10,11]. However, the extraction efficiency is always affected by a variety of factors, including the type of solvent, extraction time, solvent volume, pH, and temperature. Ensuring the extraction efficiency of active ingredients from original materials has always been the focus of food processing and natural medicine development. For a long time, researchers have been committed to revealing the relationship between yields and influencing factors through statistics and mathematical techniques [12]. Appropriate mathematical modeling provides theoretical support and technical means for optimizing extraction conditions, which can minimize the consumption of time, resources, and energy, while maximizing the quality and quantity of bioactive ingredients [13].

Response surface methodology (RSM), a collection of statistical methods, establishes the function formulas between the inputs and the response variables through a series of deterministic tests. The interaction between multiple input variables and the impact on overall efficiency can be evaluated by multiple regression and factorial design analysis in RSM, which has been extensively used to achieve optimal conditions for extraction and production in many fields, such as the chemical, pharmaceutical, and food industries [14,15]. In order to decrease experiment time and reduce the number of samples, Box–Behnken design (BBD), a response surface design approach, was developed for designing the experimental framework and analyzing the nonlinear interactions of variables. It has been extensively utilized to optimize the extraction processes of bioactive compounds, especially polysaccharides, phenolic compounds, and organic acids from various sources [16,17,18].

An artificial neural network (ANN) is a computational and mathematical modeling technique that simulates the structure of the biological central nervous system [19]. It is a better alternative to RSM due to its self-training ability in a multi-factor response optimization process, which gives the model a more reliable predictive performance. Recently, ANN has been gradually introduced into the optimization of extraction process, for instance the total phenolic, flavonoid, or/and amino acids from *Porphyra dentata*, *Nypa fruticans* Wurmb, and *Cornus officinalis* fruit [20,21,22]. As previous studies have shown, ANN, a more acceptable and superior tool, can give a more accurate prediction and effective optimization for nonlinear relation in extraction process.

Due to the initial weights and thresholds affect, the ANN model is relatively weak for global optimization and adaptability. Therefore, in order to improve the prediction performance and application scope of a model for complex problems, the model structure is usually optimized by a variety of algorithms, among which the genetic algorithm (GA) is one of the most commonly used and well-developed algorithms. It is a method to search for the optimal solution by simulating the biological evolution process, which is inspired and proposed according to natural selection and the genetic mechanisms of Darwin’s theory. It transforms the optimization process into the simulation of the crossover and mutation of chromosome genes. Compared to other algorithms, the GA can quickly obtain improved, feasible solutions with a known fitness function when facing complex combinatorial optimization, and it has been widely used in machine learning [23].

There are a growing number of studies on the pharmacology and bioactive ingredients of *P. cocos*, but to the best of our knowledge, there is an insufficient focus on optimizing the extraction parameters of various active ingredients in *P. cocos* at the same time, and there is no report on the optimization of the extraction process of *P. cocos* by ANN combined with GA. Accordingly, we designed the tests by RSM (three-factor and three-level design) and prepared the samples. The dataset was constructed by determining the ingredients (four triterpene acids and total polysaccharides and polyphenols) and antioxidant effects (2,2-diphenyl-1-picrylhydrazyl radical scavenging and total reducing capacity) of extracted samples. Furthermore, taking the experimental data from RSM run points, the ANN-GA model was developed and the best network structure was obtained. Ultimately, the optimal conditions were detected through RSM-ANN-GA and revealed the relationship between independent variables and outputs.

## 2. Materials and Methods

### 2.1. Materials

Dried *P. cocos* was purchased from a local market (Yunnan Baiyao Group Chinese Medicinal Resources Co., Ltd. Kunmin, China) in Yunnan province of China and stored at −20 °C until use. Dried *P. cocos* was ground using an electric crusher in order to pass through a 60 mesh sieve. Pachymic acid (PubChem CID: 5484385), trametenolic acid (PubChem CID: 12309443), tsugaric acid A (PubChem CID: 44422321), and dehydrotrametenolic acid (PubChem CID: 15391340) were obtained from Yuanye (Shanghai Yuanye Bio-Technology Co., Ltd. Shanghai, China). Glucose (PubChem CID: 65533), 2,2-diphenyl-1-picrylhydrazyl (DPPH), gallic acid (PubChem CID: 370), Folin–Ciocalteu reagent, DNS reagent, iron (III) chloride hexahydrate (PubChem CID: 16211236), and 2,4,6-tris(2-pyridyl)-s-triazine (TPTZ, PubChem CID: 77258) were purchased from Solarbio (Beijing Solarbio Science & Technology Co., Ltd., Beijing, China). All other reagents and solvents were of analytical grade.

### 2.2. Experimental Design of Response Surface Methodology(RSM) for Extraction

Ultrasound-assisted extraction (UAE) was conducted using an ultrasonic bath (KUNSHAN Ultrasonic Instrument CO., Ltd. KQ5200DE, Kunshan, China) at 40 kHz with an input power of 200/700 W [24,25,26]. The prepared powder (1.0 g) was put in a 50 mL centrifuge tube filled with a certain ratio of ethanol-water solution (X_1_: 25, 50, and 75%) and operation was carried out for different times (X_2_: 10, 15, and 20 min) and extraction solution volumes (X_3_: 20, 40, and 60 mL). The extraction conditions were optimized by using the Box–Behnken design (BBD). Additionally, each experiment sample was analyzed according to RSM design formulation with three replicates. The samples were centrifuged at 12,000 rpm for 20 min and extracted another two times in accordance with the above steps. All the supernatant was blended and 100 µL of the mixture was taken and diluted 100 times with 50% methanol aqueous solution. Then, the extracts were filtered through a 0.22 µm membrane filter (PVDF syringe filter, water wettable, hydrophilic, Waters Technologies Co., Ltd., SKU:186009314, Shanghai, China).

### 2.3. Determination of Contents and Activity

#### 2.3.1. Triterpene Acids Content

The four triterpene acids (PA: pachymic acid, TA: trametenolic acid, TAA: tsugaric acid A, and DA: dehydrotrametenolic acid) in the extracts were analyzed using an AB Sciex ExionLC UHPLC coupled with a QTRAP instrument. UPLC was equipped with an auto-sampler, a binary pump, a column heater, and a degasser. The samples were separated on an ACQUITY UPLC BEH C18 (100 mm × 2.1 mm; particle size, 1.7 μm). The mobile phase used was (A) water containing 0.1% formic acid and (B) acetonitrile containing 0.1% formic acid. Gradient elution was performed as follows: 20% B solvent at 0–0.50 min, 20–95% at 0.50–2.00 min, 95% at 2.00–4.50 min, 95–20% at 4.50–4.51 min. The flow rate was 0.3 mL/min at a column temperature of 40 °C. The electrospray spray ionization-quadropule ion trap (ESI-QTRAP) spectra were acquired in the negative ion mode, and the optimized parameters were as follows: ionspray voltage, 4500 V; ion source temperature, 500 °C; declustering potential, 200 eV; and collision energy, 55–58 eV. The multiple response monitoring parameters and MRM-extracted ion chromatogram of the four triterpene acids is provided in Appendix A.

#### 2.3.2. Total Polysaccharide Content (TPs)

For the analysis of total polysaccharide content, the DNS method was adopted [27]. The total polysaccharides in the extracts were hydrolyzed into reducing sugar by HCl, and the reducing sugar was reduced to a reddish-brown amino compound after co-heating with DNS reagent under alkaline conditions. After the reaction, the absorbance was determined at 540 nm using a microplate reader. Glucose was used as a standard and the results were described as mg glucose equivalent per gram (mg GLU/g) of sample powder.

#### 2.3.3. Total Phenolic Content (TPc)

Total phenolic content (TPc) was analyzed referring to a method in the literature [28]. The extracted sample (10 μL) and Folin–Ciocalteu reagent (50 μL) were mixed together in 96-well plates, then 7.5% sodium carbonate (50 μL) and distilled water (90 μL) were added to each well. After being kept at room temperature for 10 min, the absorbance was determined at 760 nm using a microplate reader with a gallic acid standard, representing mg gallic acid equivalents per gram (mg GAE/g) of sample powder.

#### 2.3.4. Antioxidant Activity

Antioxidant activity was evaluated by a 2,2-diphenyl-1-picrylhydrazyl radical scavenging assay (DPPH-SC) and a FRAP total antioxidant capacity assay (T-AOC). The free radical scavenging activity and reducing power were determined from a previous method [29].

### 2.4. Statistical Analysis

The data were expressed as means ± standard deviation. Principle component analysis (PCA) was performed by Origin 2021 and used to overview the relationship of different extracts and antioxidant activity for grouping optimization. Design Expert Software 13.0 was used to formulate the experiment cases and statistical analysis of RSM along with Neural Network Toolbox™ in MATLAB 2018a.

#### 2.4.1. RSM Modeling

ANOVA was used to analyze the statistical significance and each term of the RSM fitting model. The interaction effect of each variable on the response value was visualized by a 3D surface plot, and expressed by a modified cubic polynomial model in the following Equation (1). Eventually, the optimal extraction process was determined.
(1)$$Y=\alpha_{0}+\sum_{i=1}^{3}\alpha_{i}X_{i}+\sum_{i=1}^{3}\alpha_{\mathrm{ii}}X_{i}^{2}+\sum_{i<j=3}^{3}\alpha_{\mathrm{ij}}X_{i}X_{j}+\sum_{i<j=3}^{3}\alpha_{\mathrm{iij}}X_{i}^{2}X_{j}$$
where Y represents the dependent variables for the independent variables (X_1_–X_3_); and α_0_, α_i_, α_ii_, α_ij_, and α_iij_ are the constant coefficients of intercept, linear, quadratic, and interaction terms, respectively.

#### 2.4.2. Artificial Neural Network-Genetic Algorithm (ANN-GA) Modeling

ANN was used to explore the nonlinear correlation between independent (ethanol concentration, time, and temperature) and dependent variables (PA, TA, TAA, DA, TPs, TPc, DPPH, and FRAP). As per the work of Hee-Jeong Choi et al., the multi-layer perceptron (MLP) was constructed by input, hidden, and output layers, and the back propagation feed-forward (BPFF) model was used in the process [21]. For the network construction, all experimental data were apportioned into training (70%), testing (15%), and validation sets (15%). According to the approximation of mean square error (MSE) function, 10 hidden neurons were set in the process modeling. Each neuron was activated by using the output signals generated by the weight coefficient of the independent variables. MSE values were calculated by Equation (2), and the ANN model with minimum MSE and maximum R^2^ was selected for further optimization [30].
(2)$$\mathrm{MSE}=\frac{1}{N}\sum_{i=1}^{N}{\left(Y_{\mathrm{ANN}}-Y_{\mathrm{EXP}}\right)}^{2}$$
where Y_ANN_ and Y_EXP_ are the results from ANN prediction and experiment, respectively. The tansig function, Equation (3), was employed for pattern recognition and modeling.
(3)$$\tan\mathrm{sig}\left(n\right)=\frac{2}{1+e^{-2n}}-1$$

GA has been widely applied in multiple fields for algorithm optimization [31,32]. As previously reported, a hybrid ANN-GA could be used in ultrasonic extracting procedures [33]. After the ANN was constructed, GA was performed followed by genetic operators such as reproduction, crossover, and mutation steps until optimized results were obtained.

## 3. Result and Discussion

### 3.1. Principle Component Analysis (PCA)

PCA was conducted to visualize the correlation among all extracts (PA, TA, TAA, DA, TPs, and TPc) and antioxidant activity (DPPH-SC and T-AOC). As a powerful and common tool, PCA is able to reduce the dimensionality of the multivariate data to two or three principal components with maximized conservation of information [34]. As shown in Appendix A, the first (PC1) and second (PC2) principal components were 43.7% and 21.8%, respectively, which explain the original variance of the variables. The four triterpene acids were on the positive side of PC1, which were characterized as having high relevance (r ≥ 0.6) with each other. Additionally, there was a highly positive correlation (r ≥ 0.6) between TPs and TPc. Additionally, antioxidant activity, DPPH-SC, and T-AOC had a moderate relevance (0.4 < r < 0.6). However, the TPs and TPc had a higher correlation with the activity index [35]. Therefore, the basis had been provided for the classification of ingredients and activities. All responses could be divided into Group 1 (PA, TA, TAA, and DA) or Group 2 (TPs, TPc, and antioxidant activity) for subsequent algorithm optimization.

### 3.2. RSM Modeling

The aim was to strengthen the extraction efficiency of the four triterpene acids, polysaccharides, phenolics, and antioxidant activity from *P. cocos*. The most contributory and affected factors in the extraction process were ethanol concentration (X_1_, %), extraction time (X_2_, min), and extraction solution volume (X_3_, mL), and these were optimized by constructing a BBD formulation. Table 1 shows a comparison between response variables (PA, TA, TAA, DA, TPs, TPc, DPPH-SC, and T-AOC) and predicted responses by RSM and ANN for 17 run samples under different extraction conditions. In the design matrix, all the experimental values were quite near to the RSM- and ANN-predicted values. ANOVA was performed to evaluate RSM models for better precision. As the ANOVA results illustrate in Table 2, each model could reflect the relationship between input variables and output responses with a higher *F*-value, lower probability value (*p* < 0.05), and an insignificant lack of fit value. Moreover, the regression coefficient (R^2^) and adequate precision were also important indicators of the model fitting. In our research, R^2^ values of all responses fell within the acceptable range (R^2^ ≥ 0.74), which indicated that these models had good reliability and fit to the responses with our equations [36,37]. Adequate precision was an index of signal to noise ratio. A value of greater than four was desirable, which indicates the model could be used to navigate the design space. Our results satisfied this acceptable minimum limit [38,39]. As previous research has demonstrated, *p*-values (the lower the better), *F*-values (the higher the better), and R-squared values (ideally as close to 1.00 as possible) suggest the predictive power of a model [40]. Sushma Chakraborty et al. optimized an ultrasound-assisted extraction process for bioactive compounds from bitter gourd by response surface methodology, and all *p*-values < 0.0001, *F*-values ≥ 29.36, and R^2^ ≥ 0.9635 in RSM models, which was similar to our research [41]. In Chen Chen’s study, the F-test had a very high model *F*-value (73.85) and very low *p*-values (*p* < 0.0001), which means that there was only a 0.01% chance that a large model *F*-value could be attribute to noise [42]. The bigger the *F*-value, the smaller significance of the corresponding coefficient, which implies the model is suitable for use. In summary, the current experimental data was suitable for optimization of extract conditions. 

The interactive influence of factors (ethanol concentration, solvent volume, and extraction time) on the responses (PA, TA, TAA, DA, TPs, TPc, DPPH-SC, and T-AOC) was visually analyzed by 3D response surface plots. The four triterpene acids yield, ethanol concentration, and extractant volume had similar tendencies, indicating a gradual increase in extraction rate with more ethanol and at larger volumes. However, extraction time had a relatively smaller effect on it (Figure 1). For TPs and TPc, the effects of extraction conditions were complex. Generally, the yield increased as the extractants increased. However, it slightly decreased with an increase in time with the same ethanol concentration and an extractant volume lower than 50 mL (Figure 2). At different durations or lower solvent volumes, the content of TPs declined with a decrease in ethanol concentration then increased as the ethanol increased. However, it showed a parabolic trend, firstly increasing and then decreasing, with a higher volume of extraction solution. Additionally, the TPc content was significantly influenced by ultrasonic time at a constant ethanol concentration and solvent volume, showing the highest amount at 25% ethanol in 40 mL extractant for 30 min in the experimental data. For antioxidative activity, the impact of ethanol concentration and extraction duration on DPPH-SC and T-AOC followed similar trends. The activity was increased in a small range with an increase in ethanol content in then extract when the other factors were kept constant. Additionally, with an increasing duration, the two activity responses were first increased and then decreased with a constant ethanol concentration and extractant dosage. Moreover, as the dosage was increased with a constant ethanol content or extraction time, DPPH-SC first decreased and then increased, while T-AOC decreased under the same conditions.

Based on the response data, it is clearly shown that ethanol concentration, extraction solution volume, and extraction time exert effects on all response parameters. Appropriately increasing the ethanol concentration, solvent amount, and ultrasonic time could result in an increased yield of the four triterpene acids. However, the anti-oxidative activity (DPPH-SC and T-AOC) and bioactive components (TPs and TPc) decreased upon prolonging the ultrasonic process after reaching the maximum level. This was probably due to the degradation mechanism on exposure to powerful ultrasonic energy for a long time [43,44]. All above data revealed a different impact for all extraction ingredients and activities from the three investigate factors. Therefore, these outputs could be grouped, which was consistent with PCA analysis. This did not mean that the triterpene acids had no antioxidant activity, although they were divided into two groups according to PCA and RSM analysis. In other words, there was a weaker correlation of the four triterpenoid acids in this study with DPPH-SC and T-AOC compared with TPs and TPc. Thinzar Aung et al. also used this method to study the extract process optimization of functional components from *Porphyra dentate* [20]. The ethanol amount, time, and solvent volume were found to exert significant effects on the release of triterpene acids. This is in good agreement with previous studies of triterpene acid extraction from Corni Fructus and olives using sonication [20,45]. Moreover, the stability and extractability of total polysaccharides and total phenols at longer ultrasonic durations were in good agreement with the ultrasound-assisted extraction research in previous work [46,47]. Antioxidant function is one of the most studied biological activities of *P. cocos* and is associated with a variety of components, such as triterpene acids, TPs, and TPc [5,48,49]. Therefore, the bioactivity indexes (DPPH-SC and T-AOC) exhibited a similar trend with TPs and TPc. The more appropriate the extraction conditions, the better the extraction efficiency of antioxidants and active ingredients of *P. cocos*.

### 3.3. RSM-ANN-GA Modeling

The extraction conditions were set as ANN inputs for optimization and the data generated from RSM experimental responses was fed in the output layer. The artificial neural network architecture topology of both groups is shown in Figure 3A. All the data were randomly allocated to training, testing, and validation sets for modeling. The experimental and predicted regressions for each group in the ANN are presented in Figure 3B,E and for each output in Appendix A. The best fit was obtained after 1000 iterations by using a feed-forward and BP algorithm according to previous studies [50,51]. As shown in Figure 3C,F, GA was used to optimize the weights and thresholds of each node in ANN for an optimal network structure. The initial population was generated from the RSM-ANN model. Finally, the optimum fit and minimum sum-square error were obtained within 100 iterations. In Table 2, all RSM-ANN-GA prediction values were compared with predicted and experimental RSM data for each test condition. Then, the training process was evaluated by mean square error (MSE), Figure 3D,G. Additionally, MSE values dropped rapidly to reach a minimum within 10 epochs, which meant the best validation performance had been reached for each model and for each output in Appendix A [52]. At this stage, training was stopped, and the weights and biases were applied in processing for generation of the RSM-ANN-GA model. Obviously, the RSM-ANN-GA model with a higher coefficient of determination (R^2^) attained a better predictive power and higher predictive accuracy than the RSM model. Based on the regression analysis above, these models met with the statistical agreement between experimental data and predicted values.

ANN-GA modeling is a well-known, flexible, powerful technology. It has been widely used in many research fields with the capacity to simultaneously optimize multiple variables. Except for in the optimization of an ANN network structure, GA also was applied in optimization tasks. As reported by Lahiri and Ghanta, only scalar values can be used in objective functions, instead of the second- and/or first-order derivatives of it [53]. For maximizing outputs, the network data from RSM-ANN-GA as the objective function and the three independent variables as a matrix variable with boundary constraints is taken as follows:[25;30;20] ≤ [X_1_;X_2_;X_3_] ≤ [75;50;60]

The parameters applied in the GA optimization process were used as the default in the optimization tool of Matlab. The constrain-dependent population size was 50, and rank scaling function was carried out [54]. The stochastic uniform was selected to choose parents for the next generation based on their scaled values from the fitness scaling function [55]. A population size of 0.05× was set as the elite count guaranteed to survive to the next generation, and the default crossover fraction of 0.8 was used to produce a new generation in the reproduction process. Choosing a constraint-dependent mutation function provided genetic diversity and enabled the GA to search a broader space. The crossover function was also constraint dependent and was used to form a new individual or child for the next generation. A forward migration was adopted with 20% probability and 20 intervals. The augmented Lagrangian, an nonlinear constraint algorithm, was applied to achieve the required accuracy [56].

### 3.4. Optimum Ultrasonic Extraction Conditions

The optimization of two group responses was statistically evaluated by comparing RSM with the hybrid RSM-ANN-GA in Table 3. According to the RSM model, the best predicted yield of four triterpene acids in Group 1 was reached at 55.97% ethanol concentration, after 49.30 min, and with 60.00 mL of extraction agent with 0.9 desirability. For Group 2, 25.00% ethanol, 30.00 min, and 20.00 mL were more suitable, with 0.87 desirability. Under their own optimal conditions, the responses corresponding to PA, TA, TAA, DA, TPc, TPs, DPPH-SC, and T-AOC value were 697.92, 51.93, 184.87, 108.86 μg/g, 38.82 mg GLU/g, 319.78 μg GAE/g, 10.24%, and 1.77 μmol/g, respectively. Regarding the RSM-ANN-GA model, the optimized extraction conditions of Group 1 were 53.53% ethanol concentration, 48.64 min, and 60.00 mL, similar to the RSM model predictions. For Group 2, the optimized extraction time and volume were consistent with RSM results. The predicted output for each optimized condition exhibited an approximation for different models to be calculated. However, a higher ethanol concentration (25% to 40.49%) was required for achieving the same yield as RSM in Group 2. It could be observed that the predicted values in the RSM-ANN-GA model were lower than RSM alone, except for DPPH-SC, which was slightly higher in the RSM-ANN-GA model, but it was more credible and accurate based on the above modeling analysis [20,21].

## 4. Conclusions

In the present study, the optimization of ultrasound-assisted extraction was successfully carried out for the preparation of *P. cocos* extracts at different ethanol concentrations (25, 50, and 75%), times (30, 40, and 50 min), and extractant volumes (20, 40, and 60 mL) using two different types of statistical approaches. RSM was used to predict and optimize the global results with the minimum number of test samples, and PCA was applied to classify multiple responses reasonably. The response indicators to be optimized were PA, TA, TAA, DA, TPc, TPs, DPPH-SC, and T-AOC values. On comparing the predictive ability of the two types of mathematical modeling, the RSM-ANN-GA approach proved to be more reliable and accurate, exhibiting a higher R^2^ value than the RSM model. This research not only proposes a statistic process that follows experimental design, variable correlation analysis, modeling, model structure optimization, and prediction, but also proposes a suitable extraction technology for the preparation of antioxidant constituent and multi-component *P. cocos* extracts. Of course, it should be emphasized that the affecting factors were complex and diverse for ingredients and activities extracted from *P. cocos*, such as pH, grinding fineness, frequency of ultrasound, and even origin and species. These factors limited the application scope of the model established in this study. Moreover, qualitative and quantitative analyses of bioactive components should also be explored for further purposeful modeling and optimization.

## Figures and Tables

**Figure 1 foods-12-00619-f001:**
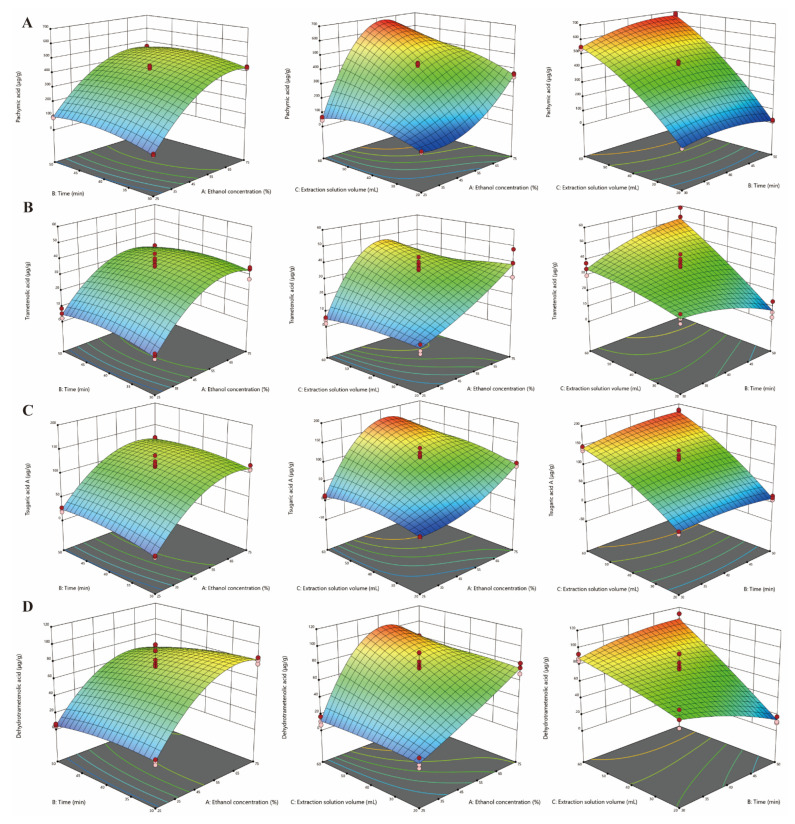
Response surface plots of ultrasound-assisted extraction conditions displaying the influence of independent variables: (**A**) pachymic acid (PA), (**B**) trametenolic acid (TA), (**C**) tsugaric acid A (TAA), (**D**) dehydrotrametenolic acid (DA).

**Figure 2 foods-12-00619-f002:**
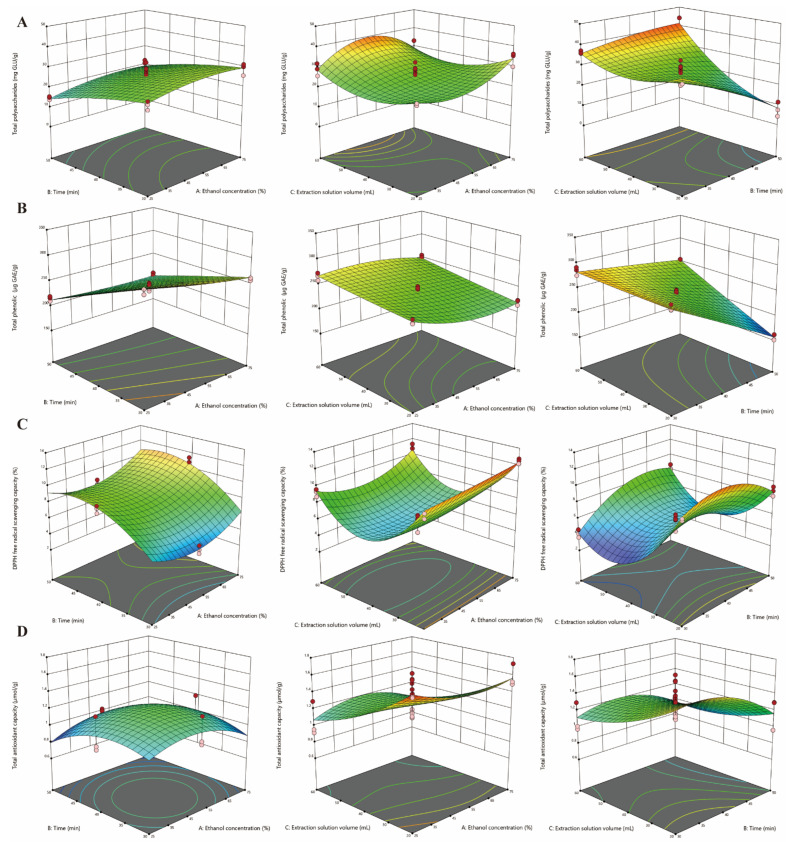
Response surface plots of ultrasound-assisted extraction conditions displaying the influence of independent variables (**A**) total polysaccharides (TPs), (**B**) total phenolic (TPc), (**C**) DPPH free radical scavenging capacity (DPPH-SC), (**D**) total antioxidant capacity (T-AOC).

**Figure 3 foods-12-00619-f003:**
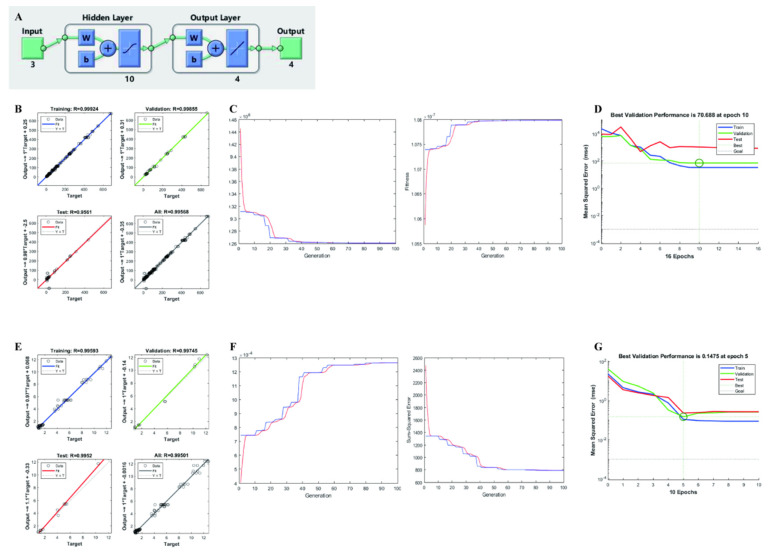
ANN modeling and training. (**A**) ANN architecture topology of both extraction conditions, (**B**) regression of experimental and predicted values in ANN model of Group 1, (**C**) GA algorithm optimization ANN model of Group 1, (**D**) ANN training performance of Group 1, (**E**) regression of experimental and predicted values in ANN model of Group 2, (**F**) GA algorithm optimization ANN model of Group 2, (**G**) ANN training performance of Group 2.

**Table 1 foods-12-00619-t001:** Summary of design matrix for investigated responses (n = 3).

Run	Variables			Responses											
	A (%)	B (min)	C (mL)	PA (μg/g)			TA (μg/g)			TAA (μg/g)			DA (μg/g)		
				Exp	Pred		Exp	Pred		Exp	Pred		Exp	Pred	
					RSM	ANN		RSM	ANN		RSM	ANN		RSM	ANN
1	50	40	40	434.57 ± 10.16	422.08	419.80	35.78 ± 4.01	34.78	34.56	109.94 ± 6.95	113.24	113.37	68.75 ± 5.12	71.17	72.89
2	50	50	20	17.54 ± 3.84	13.77	10.88	7.36 ± 4.36	7.30	13.21	13.43 ± 4.39	12.22	11.79	12.89 ± 3.26	11.39	13.69
3	75	40	20	355.47 ± 10.52	355.32	359.32	39.44 ± 7.00	38.61	43.74	94.36 ± 3.82	95.00	97.10	74.90 ± 5.06	69.26	75.55
4	75	40	60	487.26 ± 5.02	487.11	495.19	29.06 ± 5.25	28.23	25.86	119.07 ± 5.62	119.70	119.36	75.48 ± 5.57	78.51	79.15
5	50	30	20	38.00 ± 2.64	41.78	37.87	24.81 ± 2.30	24.86	24.77	20.54 ± 3.54	21.75	24.32	59.08 ± 8.18	59.58	59.28
6	25	50	40	89.34 ± 2.62	92.97	91.50	5.36 ± 2.60	4.59	5.39	22.10 ± 4.00	23.94	19.38	5.14 ± 1.12	3.66	5.19
7	50	50	60	672.10 ± 9.77	668.33	660.04	51.25 ± 6.53	51.20	42.75	179.60 ± 3.88	178.39	178.38	104.45 ± 5.52	107.96	105.11
8	50	40	40	421.83 ± 13.04	422.08	419.80	37.65 ± 4.45	34.78	34.56	117.05 ± 6.67	113.24	113.37	81.98 ± 9.02	71.17	72.89
9	50	40	40	426.13 ± 12.66	422.08	419.80	33.03 ± 4.57	34.78	34.56	106.05 ± 2.83	113.24	113.37	67.32 ± 3.80	71.17	72.89
10	25	30	40	33.57 ± 5.03	29.65	34.59	3.42 ± 1.40	2.54	3.31	7.80 ± 1.95	7.23	6.08	11.51 ± 2.46	13.71	11.51
11	50	30	60	542.10 ± 6.40	545.87	548.24	33.99 ± 3.22	34.05	32.08	142.41 ± 4.29	143.62	146.57	88.08 ± 3.44	89.57	92.62
12	25	40	20	26.51 ± 4.86	26.66	28.90	5.12 ± 2.22	5.94	2.71	6.25 ± 1.63	5.62	6.17	10.39 ± 4.81	9.03	10.39
13	50	40	40	412.60 ± 6.02	422.08	419.80	30.97 ± 3.40	34.78	34.56	111.01 ± 6.80	113.24	113.37	68.27 ± 5.37	71.17	72.89
14	75	50	40	452.91 ± 10.50	456.83	449.44	29.45 ± 1.84	30.32	31.92	118.06 ± 3.66	118.64	119.62	63.65 ± 3.92	63.11	61.84
15	50	40	40	415.27 ± 5.11	422.08	419.80	36.49 ± 2.06	34.78	34.56	122.14 ± 12.26	113.24	113.37	79.18 ± 11.13	71.17	72.89
16	25	40	60	55.72 ± 9.37	55.87	68.54	4.10 ± 1.59	4.93	4.10	12.12 ± 1.63	11.49	8.09	11.49 ± 4.24	10.13	6.62
17	75	30	40	429.31 ± 4.72	425.69	433.54	32.00 ± 3.25	32.77	32.00	111.96 ± 4.14	110.11	111.88	81.72 ± 2.99	82.87	81.47
**Run**	**Variables**			**Responses**											
	**A** **(%)**	**B** **(min)**	**C** **(mL)**	**TPs (mg GLU/g)**			**TPc (μg GAE/g)**			**DPPH-SC (%)**			**T-AOC (μmol/g)**		
				**Exp**	**Pred**		**Exp**	**Pred**		**Exp**	**Pred**		**Exp**	**Pred**	
					**RSM**	**ANN**		**RSM**	**ANN**		**RSM**	**ANN**		**RSM**	**ANN**
1	50	40	40	29.24 ± 2.54	25.61	25.61	241.14 ± 6.11	242.71	241.76	5.55 ± 0.25	5.45	5.44	1.27 ± 0.11	1.28	1.32
2	50	50	20	7.71 ± 2.93	8.36	7.79	154.71 ± 4.89	156.76	150.76	8.89 ± 0.47	8.82	8.29	1.18 ± 0.16	1.16	1.15
3	75	40	20	33.83 ± 2.80	34.02	29.89	220.61 ± 4.53	219.60	218.77	12.50 ± 0.24	12.34	12.12	1.65 ± 0.20	1.62	1.56
4	75	40	60	28.99 ± 2.92	29.18	28.98	249.83 ± 4.45	248.82	252.23	10.78 ± 0.47	10.62	10.77	1.03 ± 0.12	1.00	1.01
5	50	30	20	36.60 ± 2.20	35.95	37.00	281.69 ± 4.55	279.63	280.08	10.46 ± 0.37	10.52	10.76	1.53 ± 0.08	1.55	1.49
6	25	50	40	14.66 ± 0.69	14.19	14.23	215.27 ± 3.93	212.21	219.85	4.23 ± 0.30	4.14	4.36	0.95 ± 0.06	0.95	0.93
7	50	50	60	40.73 ± 2.52	41.38	41.30	250.96 ± 4.55	253.02	249.22	8.27 ± 0.31	8.21	11.28	0.99 ± 0.03	0.97	1.10
8	50	40	40	22.75 ± 3.10	25.61	25.61	243.32 ± 2.03	242.71	241.76	5.80 ± 0.35	5.45	5.44	1.25 ± 0.14	1.28	1.32
9	50	40	40	26.31 ± 2.34	25.61	25.61	243.49 ± 4.66	242.71	241.76	5.29 ± 0.22	5.45	5.44	1.27 ± 0.20	1.28	1.32
10	25	30	40	24.90 ± 1.59	25.74	23.98	301.98 ± 6.95	303.03	301.85	3.57 ± 0.45	3.35	3.60	1.35 ± 0.01	1.30	1.36
11	50	30	60	36.32 ± 0.58	35.67	36.33	284.16 ± 6.67	282.11	283.83	4.13 ± 0.41	4.19	3.87	1.09 ± 0.14	1.11	0.94
12	25	40	20	26.24 ± 1.41	26.05	25.25	250.68 ± 3.52	251.68	250.01	11.10 ± 0.65	11.26	11.33	1.69 ± 0.02	1.71	1.70
13	50	40	40	23.08 ± 3.45	25.61	25.61	240.52 ± 2.97	242.71	241.76	5.27 ± 0.84	5.45	5.44	1.29 ± 0.23	1.28	1.32
14	75	50	40	19.89 ± 2.30	19.05	20.72	203.07 ± 3.83	202.02	202.05	5.44 ± 0.27	5.66	5.34	0.90 ± 0.06	0.94	0.98
15	50	40	40	26.67 ± 3.11	25.61	25.61	245.10 ± 2.79	242.71	241.76	5.32 ± 0.21	5.45	5.44	1.33 ± 0.16	1.28	1.32
16	25	40	60	28.85 ± 2.50	28.67	27.43	265.78 ± 6.10	266.79	266.57	9.23 ± 0.36	9.39	9.22	1.05 ± 0.17	1.08	1.05
17	75	30	40	28.91 ± 2.39	29.37	28.84	260.09 ± 2.26	263.16	270.32	4.03 ± 0.36	4.13	4.03	1.14 ± 0.06	1.14	1.16

A: ethanol concentration (*v*/*v*, %), B: time (min), C: extraction solution volume (mL), PA: pachymic acid (μg/g), TA: trametenolic acid (μg/g), TAA: tsugaric acid A (μg/g), DA: dehydrotrametenolic acid (μg/g), TPs: total polysaccharides (mg GLU/g), TPc: total phenolic (μg GAE/g), DPPH-SC: DPPH free radical scavenging capacity (%), T-AOC: total antioxidant capacity (μmol/g), Exp: experimental value, Pred: predicted value.

**Table 2 foods-12-00619-t002:** ANOVA analysis for the response surface model.

Source	PA	TA	TAA	DA	TPs	TPc	DPPH-SC	T-AOC
Sum of Squares (model)	2.24 × 10^6^	1.06 × 10^4^	1.50 × 10^5^	5.03 × 10^4^	3.03 × 10^3^	5.47 × 10^3^	402.52	2.36
df	10	10	10	10	10	10	10	10
Mean Squares (model)	2.24 × 10^5^	1.06 × 10^3^	1.50 × 10^4^	5.03 × 10^3^	3.03 × 10^2^	5.47 × 10^2^	40.25	0.24
*F*-value (model)	2002.85	48.25	311.73	94.24	29.98	174.22	159.14	11.16
*p*-value (model)	<0.0001	<0.0001	<0.0001	<0.0001	<0.0001	<0.0001	<0.0001	<0.0001
*F*-value (Lack of Fit)	1.58	0.36	0.45	0.09	0.54	2.12	1.45	0.43
*p*-value (Lack of Fit)	0.22	0.70	0.64	0.91	0.59	0.13	0.25	0.67
Std. Dev	10.57	4.68	6.93	7.31	3.18	5.60	0.50	0.15
R^2^	0.99	0.92	0.98	0.96	0.88	0.98	0.98	0.74
Adeq precision	133.39	22.40	53.66	29.65	22.36	56.19	38.52	11.32

PA: pachymic acid (μg/g), TA: trametenolic acid (μg/g), TAA: tsugaric acid A (μg/g), DA: dehydrotrametenolic acid (μg/g), TPs: total polysaccharides (mg GLU/g), TPc: total phenolic (μg GAE/g), DPPH-SC: DPPH free radical scavenging capacity (%), T-AOC: total antioxidant capacity (μmol/g). df: degree of freedom, Std. Dev: standard deviation, R^2^: coefficient determination, Adeq precision: adequate precision.

**Table 3 foods-12-00619-t003:** Optimized extraction conditions for four triterpenoid acids according to the analysis by RSM and RSM-ANN-GA.

Variables		Group 1		Group 2	
		RSM	RSM-ANN-GA	RSM	RSM-ANN-GA
Input (process parameters)	Ethanol concentration (*v*/*v*, %)	55.97	53.53	25.00	40.49
	Time (min)	49.30	48.64	30.00	30.25
	Extraction solution volume (mL)	60.00	60.00	20.00	20.00
Output (responses)	PA (μg/g)	697.92	674.09		
	TA (μg/g)	51.93	43.10		
	TAA (μg/g)	184.87	184.02		
	DA (μg/g)	108.86	107.44		
	TPs (mg GLU/g)			38.82	35.33
	TPc (μg GAE/g)			319.78	283.73
	DPPH-SC (%)			10.24	10.58
	T-AOC (μmol/g)			1.77	1.61

## Data Availability

Data is contained within the article or Appendix A.

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
