# Peer review of "Optimization of Ultrasonic-Assisted Extraction Conditions for Bioactive Components and Antioxidant Activity of Poria cocos (Schw.) Wolf by an RSM-ANN-GA Hybrid Approach"

_foods, 2023, doi:10.3390/foods12030619_

Round 1

Reviewer 1 Report

The manuscript entitled “Optimization of ultrasonic-assisted extraction conditions for bio-active components and antioxidant activity of Poria cocos (Schw.) Wolf by RSM-ANN-GA hybrid approach” by Chen et al. intends to describe an application of RSM and RSM-ANN-GA to optimize UAE method to obtain antioxidant compounds from Poria cocos. The manuscript is in the scope of the Foods. The paper is well written and presented. However, there are major issues regarding the scientific design of the experiments that should be addressed:

The English language should be revised as often grammatical errors are encountered.

Remove highlights according to the rules of submission.

Avoid the use of abbreviations in the abstract. It is desirable to define the terms in other parts of the manuscript.

Avoid the abbreviations in the titles and subtitles.

Describe the DNS method (2.3.2. Total polysaccharide content (TPs)).

The subtitle “statistical analysis” should be included in “2. Materials and methods”, please revise numeration.

Table 1 is very difficult to read due to the formatting, please revise.

In Results and discussion, the authors should be improving the discussion of results obtained namely the results presented in Table 1 and 2.

In conclusion, use only abbreviations, the definitions were introduced in other parts of manuscript.

Revise references according Journal rules, namely should be used abbreviated journal name.

Line 36 – remove ; after Antioxidants.

Line 42 – remove end point before reference.

Line 120 – introduce reference of previous study.

Line 155 – mg glucose equivalent per gram(mg GLU/g) of sample powder.

p-value of statistical analysis should be written in italic, revise all manuscript.

Line 87 – page 11 – Reference?

Reviewer 2 Report

In its first use within a particular document, the genus is always written in full.  In subsequent uses, the genus can be abbreviated using the first initial and a period.  Example: on first use, write Poria cocos and reserve P. cocos for subsequent references. 

How did you determine that the purchased material is Poria cocos?

In 2.2. Experimental design of RSM for extraction section; 40 kHz is a low rate for extraction. Do you have a reference for this amount?
